# Free Your Mind: Emotional Expressive Flexibility Moderates the Effect of Stress on Post-Traumatic Stress Disorder Symptoms

**DOI:** 10.3390/ijms21155355

**Published:** 2020-07-28

**Authors:** Einat Levy-Gigi, Reut Donner, George A. Bonanno

**Affiliations:** 1School of Education Bar-Ilan University, Ramat-Gan 52900, Israel; reut.donner@gmail.com; 2Gonda Multidisciplinary Brain Research Center, Bar-Ilan University, Ramat-Gan 52900, Israel; 3Psychology Department, University of Haifa, Haifa 34988, Israel; 4Teachers College, Columbia University, New York, NY 10027, USA; gab38@columbia.edu

**Keywords:** stress, expressive flexibility, PTSD, servicemen

## Abstract

Servicemen are exposed to high levels of stress as part of their daily routine, however, studies which tested the relationship between stress and clinical symptoms reached inconsistent results. The present study examines the role of expressive flexibility, which was determined according to the ability to enhance or suppress either negative or positive emotional expression in conflictual situations, as a possible moderator between stress and Post-Traumatic Stress Disorder (PTSD) symptoms. A total of 82 active-duty firefighters (all men, age range = 25–66, M = 33.59, SD = 9.56, range of years in duty service = 2–41, M = 14.37, SD = 11.79), with different duty-related repeated traumatic exposure, participated in the study. We predicted and found that firefighters with low, but not high, expressive flexibility showed a significant positive correlation between duty-related traumatic exposure and PTSD symptomology (*t*(81) = 3.85, *p* < 0.001). Hence, the greater the exposure the higher level of symptoms they exhibited. In addition, we found a difference between the moderating roles of suppressing positive and negative emotional expression, as high but not low, ability to suppress the expression of negative emotions (*t*(81) = 1.76, *p* > 0.05), as low but not high, ability to suppress the expression of positive emotions (*t*(81) = 1.6, *p* > 0.05), served as a protective factor in buffering the deleterious effect of repeated traumatic exposure. The results provide a pivotal support for the growing body of evidence that a flexible emotional profile is an adaptive one, in dealing with negative life events. However, while there is a need to update behavior, the direction of the adaptive update may differ as a function of valance.

## 1. Introduction

The relationship between repeated exposure to trauma and Post-Traumatic Stress Disorder (PTSD) remains unclear, as studies with first responders—who are frequently exposed to traumatic events as part of their occupational routine—have shown inconsistent results. Some studies report significant positive correlation between duty-related traumatic events and PTSD, i.e., the more traumatic events one is exposed to, the higher level of PTSD symptomology, e.g., [1,2]; yet others show no direct effect [3,4,5]. These mixed findings suggest that other variables likely moderate the relationship between trauma exposure and PTSD. Previous studies reveal that emotional and cognitive flexibility may play an important role in this relationship [6,7]. The current study aims to expand these finding by focusing on the moderating role of flexibility in the regulation of emotional expression.

Emotion regulation refers to one’s ability to manage his or her emotional experience, when faced with a given situation [8]. Researchers within the field of emotion regulation differentiate between two main types of regulatory strategies: one, disengagement strategies such as distraction, in which blocking an emotional stimulus from capturing selective attention and thus preventing further cognitive processing; two, engaging strategies such as reappraisal which involves allowing elaborated cognitive processing and providing semantic meaning to the stimulus. Past studies have practiced a categorical approach, which characterized distraction as always maladaptive, and reappraisal as always adaptive, in dealing with all negative situations [9,10,11,12,13]. A more recent approach offers a person–situation interactionist model, which claims there is no “right” or “wrong” regulation strategy; different strategies predict different emotional outcomes, depending on context [8,13]. Specifically, several studies have shown it is better to use reappraisal when faced with low-intensity negative contexts (e.g., when watching a picture of two sad people, use reappraisal by thinking that they support each other and soon would feel better), and distraction when faced with highly aversive situations (use distraction by thinking of yellow triangles when watching a picture of a dead body) [6,8]. Based on this work, investigators have argued that regulatory flexibility, i.e., the ability to choose and apply a proper emotion regulation strategy in a manner that corresponds with contextual demands, plays a crucial role in the successful adaptation to negative events [14].

In the current study, we focused on a unique population of active-duty firefighters. Firefighters are exposed to an annual average of hundreds of duty-related traumatic events, including buildings and factory fires, car accidents and rescuing trapped people (see [6] for estimated number of exposures to different traumatic events in this population). Hence, they represent individuals well with repeated traumatic exposure. A recent study with active-duty firefighters demonstrated that one aspect of regulatory flexibility, related to strategy choice, moderated the relationship between repeated traumatic exposure and PTSD [6]. In this study, participants watched a series of negative computer images, with low-to-high emotional intensity. After watching each image, participants were asked to choose and apply one of two regulation strategies (distraction or reappraisal) in order to cope with the emotions triggered by the image. Participants with high regulatory choice flexibility, who successfully alternated between reappraisal in the low intensity images and distraction in the high intensity images, showed no correlation between the amount of duty-related traumatic events they experienced, and the amount of PTSD symptoms presented. However, participants with low regulatory choice flexibility, who favored only one regulation strategy or alternated between reappraisal and distraction with no regard to the contextual intensity, showed increased levels of PTSD symptomology across exposure.

In addition to the emotional experience, and the ability to regulate it, emotion theorists refer to another element of the “emotion phenomena”: Emotional expression, one’s ability to enhance or suppress an outward emotional reaction, i.e., display more or less emotion than one actually feels, respectively [14]. Previous studies have identified situations in which being highly emotionally expressive was beneficiary—for example, when trying to develop and maintain social interactions [15,16]. Others have found situations in which it is best to be more discrete [17]—for example, during argument mediation (for a full review, see [14]). Expressive flexibility refers to the ability to alternate between enhancement and suppression, in a manner that corresponds with contextual demands. Similar to the person–situation interactionist model discussed earlier, which emphasizes the importance of being flexible and in accord with situational demands when regulating one’s emotional experience, growing evidence suggests the importance of expressive flexibility—as opposed to favoring either the enhancement or the suppression of emotional expression—in successful adjustment to negative events [14,18].

For example, Bonanno et al. [14] tested expressive flexibility among New York students soon after the 9/11 attacks and found it could predict changes in their emotional health and distress levels throughout a two-year time period. In this study, participants were asked to view a set of pleasant and unpleasant emotion-evoking images on a computer monitor. For each image, participants were instructed to either enhance their emotional reactions, suppress their emotion reactions, or simply view the images as they would normally. Facial expressiveness was subsequently coded from videotapes. Expressive enhancement and expressive suppression abilities were then determined for each participant relative to that participant’s own level of expressiveness in the normal view condition. An overall expressive flexibility score was also calculated, representing the ability to both enhance and suppress emotional expression. Both the individual enhancement and suppression ability scores and the overall expressive flexibility score predicted better adjustment, i.e., reduced distress, two years after the 9/11 attacks. A limitation of this study, however, was that it did not examine differences in trauma exposure among participants.

There are also more general limitations of the expressive flexibility paradigm [19]. First, being a laboratory task, it cannot easily be employed for prospective field research. Second, the requirement that participants express or suppress emotion while sitting alone at a computer lacks ecological validity. In order to accommodate these limitations, Burton and Bonanno [19] developed the Flexible Regulation of Emotional Expression (FREE) scale as an easy to use questionnaire measure of expressive flexibility. Crucially, the FREE scale showed good validity, including correlation with expressive flexibility performance in the experimental paradigm.

The current study used the FREE scale to assess expressive flexibility among Israeli firefighters and examine the role of expressive flexibility in the elusive relationship between repeated traumatic exposure and PTSD symptomology. Based on previous findings, our main hypothesis was that expressive flexibility would moderate the relationship between trauma exposure and PTSD. Specifically, we predicted that among individuals with low, but not high, expressive flexibility, repeated traumatic exposure would be related to increased levels of PTSD symptomology. Our second and third hypotheses were that the abilities to flexibly enhance or suppress emotional expression would separately function as moderators, in the same way.

## 2. Method

### 2.1. Participants

A total of 82 active-duty firefighters (all men, age range = 25–66, M = 33.59, SD = 9.56, 34.1% single; 56.1% married; 9.8% divorced; M years of education = 12.33, SD = 1.01; range years in service 2–41, mean years = 14.37, SD = 11.79), who serve in fire stations in central Israel, volunteered to participate in this cross-sectional study. All the participants hold operational positions and hence they keep a regular fitness routine. Since the vast majority of Israeli firefighters are men, no woman participated in the study. Exclusion criteria for all participants were current or past diagnosis of Axis I and II psychopathology other than PTSD; risk of suicidal/homicidal ideation; any substance dependence or abuse within the past 6 months (excluding nicotine); a history of concussion or other clinically significant head injury including loss of consciousness for over 10 min; or a history of neurological disorders such as epilepsy, multiple sclerosis, stroke, or encephalitis; acute hearing loss or color blindness. The study was conducted in accordance with the Helsinki declaration and was approved by the ethics committee of the Fire and Rescue Service.

### 2.2. Measures

Duty-related repeated traumatic exposure—repeated traumatic exposure was measured by a self-report exposure questionnaire. In this questionnaire, participants were asked to rate the frequency in which they were personally exposed to 16 different life-threatening, duty-related events during an average year of service (e.g., building fires, car accidents, gas leaks), on a scale of 1 (“was not exposed to at all”) to 6 (“was exposed to on a daily basis”). The number of traumatic duty-related events on an average year (range = 26–66, M = 43.62, SD = 9.57) was then multiplied by each participant’s number of years as an active firefighter (range = 2–41 years, M = 14.37, SD = 11.79), in order to obtain an adequate estimation of cumulative repeated trauma throughout the service (range = 52–2440, M = 671.34, SD = 632.27).

The Flexible Regulation of Emotional Expression (FREE) scale—in the FREE scale [19], participants were instructed to indicate how well they would be able to be more or less expressive, when feeling negative or positive emotions and faced with given, hypothetical social scenarios, on a scale of 1 (“Unable”) to 6 (“Very able”). The scale contains 16 items, which are divided into two subscales: enhancement and suppression. Each subscale is divided into two groups: positive emotional expression, and negative emotional expression. Four items refer to the ability to enhance positive emotional expression (e.g., “You receive a gift from a family member but it’s a shirt you dislike”). Four items refer to the ability to enhance negative emotional expression (e.g., “Your friend is telling you about what a terrible day they had”). Four items refer to the ability to suppress positive emotional expression (e.g., “You are in a training session and you see an accidentally funny typo in the presenter’s slideshow”). Four items refer to the ability to suppress negative emotional expression (e.g., “You have just heard about the death of a close relative right before an important work meeting”). The items of each group were summed into four different scores: positive enhancement, negative enhancement, positive suppression and negative suppression. Both positive and negative enhancement scores were summed into an “enhancement score” (from here on enhancement flexibility); both positive and negative suppression scores were into a “suppression score” (from here on suppression flexibility). Expressive flexibility was calculated by subtracting the absolute value of the average difference between enhancement and suppression scores, from the average sum of the two scores (range = 4.75–10.74, M = 7.77, SD = 1.33). Higher scores indicate high expressive flexibility; lower scores indicate low expressive flexibility.

The PTSD checklist for DSM-5 (PCL-5) questionnaire [20]—this was administered to assess levels of PTSD symptomology (Weathers et al., 2013). In this questionnaire, participants were asked to indicate how much they had been bothered by different problems in the last month, on a scale of 1 (“Not at all”) to 5 (“Extremely”). The PCL-5 questionnaire contains 21 items, which were summed in order to get a total symptom severity score (range = 21–68, M = 29.24, SD = 10.78). 

### 2.3. Procedure

All participants provided a written informed consent at the beginning of the experiment after the nature of the procedure had been fully explained. Firefighters were approached in their fire station, during their shift, and were asked to fill out our three questionnaires in the following order: duty-related traumatic exposure, FREE scale and PCL-5. Firefighter trainees were approached in the National Fire and Rescue Academy in Rishon LeZion during their final week of an advanced training course, and were asked to fill out two questionnaires—FREE scale and PCL-5—in the same order. Each participant performed the experiment individually.

### 2.4. Data Analysis 

Data were analyzed with SPSS (version 25) software. In order to test our hypotheses, regarding the role of expressive flexibility, enhancement flexibility and suppression flexibility in the relationship between repeated traumatic exposure and PTSD symptomology, Hayes’s [21] PROCESS macro for moderator analysis was employed, using 5000 bootstrap resampling for calculation of confidence intervals (Model 1; for the advantages of using this macro see [22]). Duty-related repeated traumatic exposure was treated as the independent variable and PTSD symptomology was treated as the outcome. Expressive flexibility, enhancement flexibility and suppression flexibility were treated, separately, as moderators, in accordance with our three hypotheses.

## 3. Results

Zero-order correlations between all measures are reported in Table 1. We found a significant positive correlation between duty-related repeated traumatic exposure and PTSD symptomology, a significant negative correlation between enhancement flexibility and PTSD symptomology and a significant negative correlation between expressive flexibility and PTSD symptomology. We found no correlation between enhancement flexibility and suppression flexibility, indicating their independence. Both flexibility variables were positively correlated with expressive flexibility, confirming that higher abilities to enhance or suppress an emotional expression are reflected in a higher expressive flexibility score.

First, we examined our main prediction with expressive flexibility as a possible moderator, in the relationship between repeated traumatic exposure and PTSD symptomology. The estimated coefficients of the main findings and their significance levels are described in Table 2. The general model was significant, *R*^2^ = 0.25, *F*(3,78) = 8.56, *p* < 0.001. Consistent with our hypothesis, there was a significant interaction between duty-related repeated traumatic exposure and expressive flexibility, which accounted for an additional 3.7% of the variance above. To interpret the interactive effect on PTSD symptomology, we computed bootstrapping confidence intervals (95%) evaluating the magnitude of the relationship between repeated traumatic exposure and PTSD symptoms for individuals with low (−1 SD) and high expressive flexibility (+1 SD). As expected, the results revealed a significant positive relationship between duty-related repeated traumatic exposure and PTSD symptomology for individuals with low expressive flexibility, *β =* 0.012, 95% CI = 0.006, 0.018, *t*(81) = 3.85, *p* < 0.001. However, no such relationship was found among individuals with high expressive flexibility, *β* = 0.001, 95% CI = −0.01, 0.01, *t*(81) = 0.12, *p >* 0.05. These results indicate that among low (but not high) expressive flexibility individuals, an increase in duty-related traumatic exposure is associated with enhanced PTSD symptomology.

The findings regarding the moderating role of enhancement flexibility did not support our second hypothesis. While the general model was significant, *R*^2^ = 0.22, *F*(3,78) = 7.24, *p* < 0.001, the interaction between duty-related repeated traumatic exposure and enhancement flexibility was not significant, *t*(81) = −0.56, *p >* 0.05. Additionally, the results did not support our third hypothesis, regarding the role of suppression flexibility as a possible moderator. While the general model was significant, *R*^2^ = 0.19, *F*(3,78) = 6.22, *p* < 0.001, the interaction between duty-related repeated traumatic exposure and suppression flexibility was not significant, *t*(81) = −1, *p >* 0.05.

To further examine the role of the different components of expressive flexibility in the elusive relationship between duty-related repeated traumatic exposure and PTSD symptomology, we separated between the expression of positive and negative emotions and tested four different variables as possible moderators: negative enhancement, positive enhancement (i.e., the abilities to enhance the expression of positive and negative emotions, respectively), negative suppression and positive suppression (i.e., the ability to suppress the expression of positive and negative emotions, respectively). Both positive enhancement and negative enhancement yielded significant general models, however, the interactions between repeated traumatic exposure and the two variables were not significant.

The findings regarding *negative suppression* as a possible moderator are reported in Table 3. The general model was significant, *R*^2^ = 0.26, *F*(3,78) = 9, *p* < 0.001, and there was a significant interaction between repeated exposure and negative suppression which accounted for an additional 4.3% of the variance above. Further examination of this interactive effect revealed a significant positive relationship between repeated duty-related traumatic exposure and PTSD symptomology for individuals with a low ability to suppress the expression of negative emotions, *β* = 0.012, 95% CI = 0.007, 0.018, *t*(81) = 4.35, *p* < 0.001. However, no such relationship was found among individuals with a high ability to suppress the expression of negative emotions, *β* = 0.004, 95% CI = −0.001, 0.01, *t*(81) = 1.76, *p >* 0.05. These results are consistent with our general assumption, as they indicate that among low (but not high) expressive flexibility individuals—i.e., individuals with a low ability to suppress the expression of negative emotions—an increase in duty-related traumatic exposure is associated with enhanced PTSD symptomatology (as described in Figure 1).

The findings regarding positive suppression as a possible moderator are also reported in Table 3. The general model was significant, *R*^2^ = 0.23, *F*(3,78) = 7.79, *p* < 0.001, and there was a significant interaction between repeated exposure and positive suppression which accounted for an additional 6.4% of the variance above. Further examination of this interactive effect revealed a significant positive relationship between repeated duty-related traumatic exposure and PTSD symptomology for individuals with a high ability to suppress the expression of positive emotions, *β* = 0.02, 95% CI = 0.01, 0.02, *t*(81) = 4.3, *p* < 0.001. However, no such relationship was found among individuals with a low ability to suppress the expression of positive emotions, *β* = 0.004, 95% CI = −0.001, 0.01, *t*(81) = 1.6, *p >* 0.05. These results are inconsistent with our general assumption, as they indicate that among high (but not low) expressive flexibility individuals—i.e., individuals with a high ability to suppress the expression of positive emotions—an increase in duty-related traumatic exposure is associated with enhanced PTSD symptomatology (as described in Figure 2).

This pattern of results indicates that the abilities to suppress the expression of negative and positive emotions have opposing roles in the relationship between repeated traumatic exposure and PTSD symptomology.

## 4. Discussion

The aim of the current study was to examine the role of expressive flexibility—i.e., the ability to enhance and suppress an outward emotional expression—in the relationship between repeated exposure to traumatic events and PTSD symptomology. In accordance with our prediction, we found that expressive flexibility moderates the relationship between repeated traumatic exposure and PTSD symptoms. Specifically, while individuals with low expressive flexibility showed a positive association between level of repeated exposure and PTSD symptoms, no such relationship was found in individuals with high expressive flexibility. These results are in line with previous studies which showed that high expressive flexibility predicts emotional adaptation following both exposures to general trauma [14,23] or loss of partner [24]. Moreover, it complements and expands previous results on active duty firefighters, which show a similar effect of regulatory choice flexibility [6]. Finally, the results provide additional support for the importance of emotional flexibility in psychological well-being, and—for the first time, to our knowledge—provide specific evidence for the crucial role of expressive flexibility in the successful adaptation to repeated traumatic exposure.

While these results may tell a simple and clear story, focusing on the specific abilities to suppress and enhance emotions, revealed a more complex picture. First, we found that contrary to our prediction, the ability to display more emotion, neither positive nor negative, did not buffer the deleterious effects of repeated exposure to trauma. These results may question the traditional approach, which assumes that expressing emotion is therapeutic and beneficiary, and hence may represent an adaptive way to cope with trauma. Indeed, actively disclosing emotions regarding upsetting life events—whether to a close friend, a therapist or even writing about the experience—is often associated with an immediate emotional relief, and beneficial to mental and physical health [25,26,27,28,29]. Abstaining from emotional expression of past or recent adverse experiences, on the other hand, is commonly linked to poorer health [27,30], increased emotional and physical arousal [30,31] and even to cognitive impairments [32]. However, more recent studies present a different approach. Kennedy-Moore and Watson [26] have pointed out a paradox in expressing negative emotions, suggesting that while disclosure of emotions is an important part of a successful therapeutic process, it may also indicate severe emotional distress, and even serves as a warning sign. Moreover, the relationship between emotional expression and psychological well-being seems more complicated with possible mediators such as different personality traits and coping styles, initial physical state and the level of social support [33,34]. Finally, it was suggested that the personal and social elements of emotional expression play a crucial role in this relationship. Specifically, it was found that sharing emotions is most helpful in reducing distress when it is done in a supportive social environment. However, if those in need or their potential supporters feel uncomfortable communicating emotions, or when the content of the expressed emotion is too intense or overwhelming, the recipients might react in a diminishing or minimizing manner and the person in need would end up feeling dismissed or rejected. Taken together, it is possible that in our study, the intense lifestyle of male, active-duty firefighters, who are repeatedly exposed to overwhelming sights, addresses certain personal coping styles and/or social communication habits in which expressing emotions is not common nor considered as worthwhile. Alternatively, it is possible that while the majority of previous research revealed similar results in firefighters and other servicemen who are repeatedly exposed to trauma [35,36,37,38], future studies may aim to test the moderating role of expressive flexibility in the relationship between trauma exposure and PTSD symptoms in other populations including not only soldiers and police officers, but also men and women civilians who live in conflict zones and experience repeated trauma.

When we tested the general ability to suppress emotional expression, we found no significant effect on the relationship between repeated exposure to trauma and PTSD symptoms. However, further examination revealed an interesting difference between the role of suppressing negative and positive emotions. Specifically, both the ability to suppress negative emotions and the tendency to allow and avoid suppression of positive emotions served as protective factors against the effects of repeated traumatic exposure. These findings may suggest that not only suppressing negative emotions but also allowing positive emotions plays an important role in coping with trauma. Indeed, there is growing evidence for the unique role of positive emotions in stress-related situations. For example, a recent study suggests that it is the lack of positive experiences, and not the excess of negative emotions, that differentiates between individuals with social anxiety disorder and healthy controls [39]. Folkman [40] refers in her review to the important function of positive emotions in restoring physical and psychological coping resources when facing a stressful event. Similarly, the Broaden-and-Build Theory [41,42] emphasizes the crucial role of positive emotions in the down-regulation of sympathetic arousal caused by stress, and the broadening of important coping functions such as attention, creativity and flexible thinking. With that being said, it should be noted that flexibility is the capacity to use the best mechanism, depending on the situation. Thus, enhancement might be best in one situation, or at one point in time in dealing with the situation, while suppression might be best later in the same situation or in another situation. Hence, while some aspects are more related to reduced PTSD symptoms, adaptive behavior is the ability to use enhancement and suppression when they are most appropriate.

The current study has several limitations. First, we did not address the possible relationship between emotional experience and emotional expression. Previous studies which tried to answer this question have yielded inconsistent results. For example, Gross and Levenson [43] showed heightened sympathetic activity among participants who were asked to suppress all outward signs of emotional expression. Interestingly, despite this increase in physiological stress levels, participants did not report an increase in their negative emotional experience. Moreover, in some situations the expressing of negative emotions was linked to a relief from negative emotional experience [33], although this was not the case for all the participants who expressed their aversive feelings. [44,45]. Future studies may aim to test the relationship between expressive flexibility and regulatory flexibility in conditions of repeated traumatic exposure. A positive correlation, for example, would indicate that an improvement in one aspect of emotional flexibility could benefit the other. Both expressive and regulatory flexibility seem to play a crucial role in buffering the effect of traumatic exposure. Hence, a better understanding of the relationship between the two could have important theoretical and therapeutic implications. Another limitation relates to the fact that we tested participants only at one time-point. While such a design serves as a proof of concept and allows to reach important conclusions regarding the associations between the variables, future studies may aim to use a longitudinal approach, in order to base a causal relationship.

The results of the present study may serve to develop interventions which aim to minimize the deleterious effect of repeated trauma both before and during active service. Specifically, it is possible that if first responders will acquire and practice adaptive expressive flexibility skills, it will help them to better cope with stressful situations and to invigorate through positive experiences.

To summarize, the current study reveals that a flexible emotional expressive profile is important when dealing with negative life events. Most importantly, among individuals who are repeatedly exposed to trauma, a resilient expressive profile is not necessarily having high abilities to suppress and enhance positive and negative emotional expressions, but adjusting these abilities in correspondence with the complex context of their lives.

## Figures and Tables

**Figure 1 ijms-21-05355-f001:**
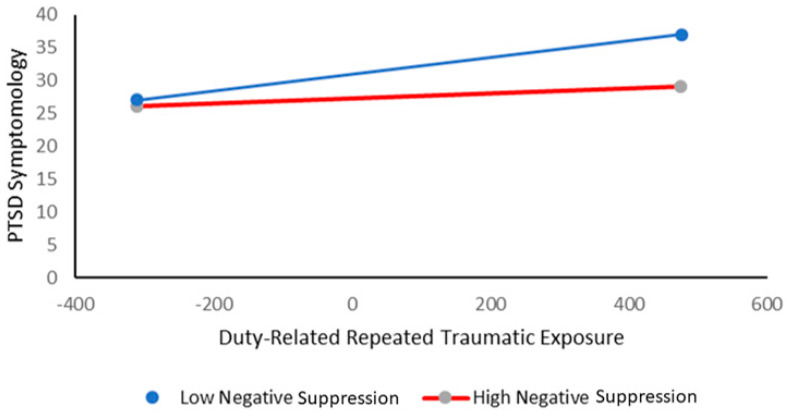
Post-Traumatic Stress Disorder (PTSD) symptomology as a function of duty-related repeated traumatic exposure and negative suppression.

**Figure 2 ijms-21-05355-f002:**
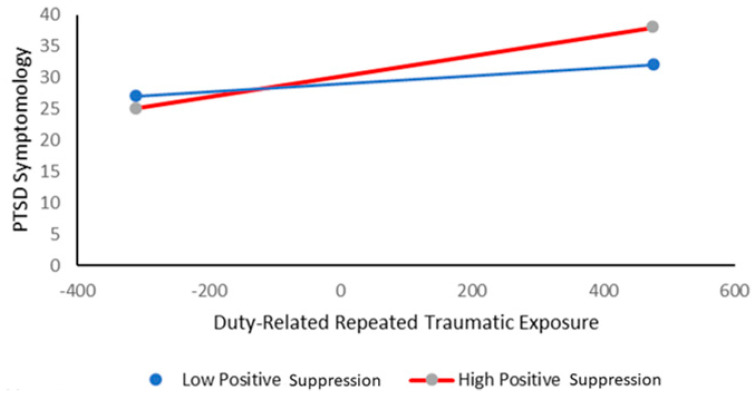
Post-Traumatic Stress Disorder (PTSD) symptomology as a function of duty-related repeated traumatic exposure and positive suppression.

**Table 1 ijms-21-05355-t001:** Zero-order correlation between enhancement flexibility, suppression flexibility, expressive flexibility, PTSD symptomology and duty-related repeated traumatic exposure.

Variable	1	2	3	4	5
1. Enhancement flexibility	1	0.203	0.69 **	−0.29 **	−0.17
2. Suppression flexibility	0.203	1	0.67 **	−0.20	−0.18
3. Expressive flexibility	0.69 **	0.67 **	1	−0.29 **	−0.20
4. PTSD symptomology	−0.29 **	−0.20	−0.29 **	1	0.41 **
5. Duty-related repeated traumatic exposure	−0.17	−0.18	−0.20	0.41 **	1

** *p* < 0.01. Post-Traumatic Stress Disorder (PTSD).

**Table 2 ijms-21-05355-t002:** Estimated coefficients, standard errors, and 95% confidence intervals for independent variable and moderator in the model predicting PTSD symptomology.

Variable	*β*	SE	*t* Value	95% CI	
				Low	High
Duty-related repeated traumatic exposure	0.006	0.002	3 **	0.002	0.01
Expressive flexibility	−2.26	0.85	−2.65 **	−3.96	−0.56
Interaction	−0.004	0.002	−1.98 *	−0.001	0

*β* = Unstandardized Estimated Coefficients; SE = Standard Error; CI = Confidence Intervals. * *p* = 0.05. ** *p* < 0.01.

**Table 3 ijms-21-05355-t003:** Estimated coefficients, standard errors, and 95% confidence intervals for independent variable and moderators in the model predicting PTSD symptomology.

**Negative Suppression as a Moderator**					
**Variable**	***β***	**SE**	***t* Value**	**95% CI**	
				**Low**	**High**
Duty-related repeated traumatic exposure	0.001	0.002	4.21 **	0.004	0.012
Negative suppression	−0.55	0.26	−2.14 *	−1.06	−0.04
Interaction	−0.001	0.000	−2.12 *	−0.002	−0.000
**Positive Suppression as a Moderator**					
**Variable**	***β***	**SE**	***t* Value**	**95% CI**	
				**Low**	**High**
Duty-related repeated traumatic exposure	0.01	0.002.	4.77 **	0.006	0.015
Positive suppression	0.23	0.30	0.76	−0.37	0.83
Interaction	0.001	0.00	2.54 *	0.000	0.003

*β* = Unstandardized Estimated Coefficients; SE = Standard Error; CI = Confidence Intervals. * *p* < 0.05. ** *p* < 0.01.

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
