# Peer review of "Free Your Mind: Emotional Expressive Flexibility Moderates the Effect of Stress on Post-Traumatic Stress Disorder Symptoms"

_ijms, 2020, doi:10.3390/ijms21155355_

Round 1

Reviewer 1 Report

The manuscript submitted by Einat Levy-Gigi is very interesting but they have some problems that they should solve.

Change PTSD by Post-traumatic stress disorder in the title

Abstract:

Explain in the background why is necessary your study.

Include Post-traumatic stress disorder (PTSD)

Include demographic data of firefighters (sex, age, ….)

What have you used to determine Emotional Expressive Flexibility?

Include numbers into results section

I do not understand your conclusion. Rewrite, please

Introduction

Use Post-traumatic stress disorder (PTSD) firs time

Use adequately the journal reference style. I think this is in [number] and not as APA.

why have you used firefighters and not for example police?

Methods

Participants

Males or females?

How long experience? This data may be could be very interesting.

Procedure

Ethic committee? Helsinki treaty?

Data analysis

Wasn't it better to have used a linear regression?

What was your limit of significance? P<0.05?

Results

Could the results not be presented with tables? It would be much easier to understand them.

Discussion

The discussion is well carried out, although I still have the doubt of knowing why they have used firefighters and not other groups.

Likewise, after the conclusion it would include a section of practical application. That is, what are these results for?

Author Response

Please find a detailed response in the attached file

Reviewer 2 Report

  • Please provide the appreciated reader with one example of the "reappraisal in the low intensity images and distraction in the high intensity images" (p.2)
  • Methods: Do the authors assume that axis II(!) psychopathology does not interfere with their results?
  • Methods: "any substance dependence or abuse within the past 6 months": including nicotine?!
  • Methods: education and relationship status of participants? BMI?
  • Methods: To my knowledge, the PCL-5 is a 20(!)-item self-report measure that assesses the 20 DSM-5 symptoms of PTSD.
  • Please explain this sentence in more detail:
    "However, in situations which either those in need or the social environment around them feel comfortable communicating emotions, or when the content of the expressed emotion is too intense or overwhelming, the recipients might react in a diminishing or minimizing manner and the expresser would end up feeling dismissed or rejected"
  • "It might be, that the intense life-style of male, active-duty firefighters, who are repeatedly exposed to overwhelming sights, dictates certain personal coping styles and/or social communication patterns in which being more emotionally expressive is not helpful" - or perhaps part of a certain type of male socialisation?!
  • "This finding might also seem surprising, since it is inconsistent
    with our main hypothesis" - No, I wasn´t surprised - and a result is not per se surprising for the audience when it is in conflict with an a priori hypothesis of some authors of a manuscript!
  • The discussion section in general shoud be revised and the results of the study should´t - not least because of methodological flaws - be exaggerated.

Author Response

Please find a detailed response to the comments attached.

Round 2

Reviewer 1 Report

The article has improved a lot and the authors have adequately answered my questions.